# MIST
## MULTIPLE INSTANCE SPATIAL TRANSFORMERS

## ABSTRACT

We propose a deep network that can be trained to tackle image reconstruction and classification problems that involve detection of multiple object instances, *without* any supervision regarding their whereabouts. The network learns to extract the most significant K patches, and feeds these patches to a task-specific network – e.g., auto-encoder or classifier – to solve a domain specific problem. The challenge in training such a network is the non-differentiable top-$K$ selection process. To address this issue, we lift the training optimization problem by treating the result of top-$K$ selection as a slack variable, resulting in a simple, yet effective, multi-stage training. Our method is able to learn to detect recurring structures in the training dataset by learning to reconstruct images. It can also learn to localize structures when only knowledge on the occurrence of the object is provided, and in doing so it outperforms the state-of-the-art.

## 1 INTRODUCTION

Finding and processing multiple instances of characteristic entities in a scene is core to many computer vision applications, including object detection (Ren et al., 2015; He et al., 2017; Redmon & Farhadi, 2017), pedestrian detection (Dollár et al., 2012; Sewart & Andriluka, 2016; Zhang et al., 2018a), and keypoint localization (Lowe, 2004; Bay et al., 2008). In traditional vision pipelines, a common approach to localizing entities is to select the top-K responses in a heatmap and use their locations (Lowe, 2004; Bay et al., 2008; Felzenszwalb et al., 2010). However, this type of approach does not provide a gradient with respect to the heatmap, and cannot be directly integrated into neural network-based systems. To overcome this challenge, previous work proposed to use grids (Redmon et al., 2016; He et al., 2017; Detone et al., 2018) to simplify the formulation by isolating each instance (Yi et al., 2016), or to optimize over multiple branches (Ono et al., 2018). While effective, these approaches require additional supervision to localize instances, and do not generalize well outside their intended application domain. Other formulations, such as sequential attention (Ba et al., 2015; Gregor et al., 2015; Eslami et al., 2015) and channel-wise approaches (Zhang et al., 2018c) are problematic to apply when the number of instances of the same object is large.

Here, we introduce a novel way to approach this problem, which we term *Multiple Instance Spatial Transformer*, or *MIST* for brevity. As illustrated in Figure 1 for the image synthesis task, given an image, we first compute a heatmap via a deep network whose local maxima correspond to locations of interest. From this heatmap, we gather the parameters of the top-$K$ local maxima, and then extract the corresponding collection of image patches via an image sampling process. We process each patch independently with a task-specific network, and aggregate the network's output across patches.

Training a pipeline that includes a non-differentiable selection/gather operation is non-trivial. To solve this problem we propose to lift the problem to a higher dimensional one by treating the parameters defining the interest points as slack variables, and introduce a hard constraint that they must correspond to the output that the heatmap network gives. This constraint is realized by introducing an auxiliary function that creates a heatmap given a set of interest point parameters. We then solve for the relaxed version of this problem, where the hard constraint is turned into a soft one, and the slack variables are also optimized within the training process. Critically, our training strategy allows the network to incorporate both non-maximum suppression and top-K selection. We evaluate the performance of our approach for ① the problem of recovering the basis functions that created a given texture,

② classification of handwritten digits in cluttered scenes, and ③ recognition of house numbers in real-world environments. In summary, in this paper we:

- introduce the MIST framework for weakly-supervised multi-instance visual learning;
- propose an end-to-end training method that allows the use of top-K selection;
- show that our framework can reconstruct images as parts, as well as detect/classify instances without any location supervision.

## 2 RELATED WORK

Attention models and the use of localized information have been actively investigated in the literature. Some examples include discriminative tasks such as fine-grained classification (Sun et al., 2018), pedestrian detection (Zhang et al., 2018a), and generative ones such as image synthesis from natural language (Johnson et al., 2018). They have also been studied in the context of more traditional Multiple Instance Learning (MIL) setup (Ilse et al., 2018). We now discuss a selection of representative works, and classify them according to how they deal with multiple instances.

**Grid-based methods.** Since the introduction of Region Proposal Networks (RPN) (Ren et al., 2015), grid-based strategies have been used for dense image captioning (Johnson et al., 2016), instance segmentation (He et al., 2017), keypoint detection (Georgakis et al., 2018), and multi-instance object detection (Redmon & Farhadi, 2017). Recent improvements to RPNs attempt to learn the concept of a generic object covering multiple classes (Singh et al., 2018), and to model multi-scale information (Chao et al., 2018). The multiple transformation corresponding to separate instances can also be densely regressed via Instance Spatial Transformers (Wang et al., 2018), which removes the need to identify discrete instance early in the network. However, all these methods are fully supervised, requiring both class *labels* and object *locations* for training.

**Heatmap-based methods.** Heatmap-based methods have recently gained interest to detect features (Yi et al., 2016; Ono et al., 2018; Detone et al., 2018), find landmarks (Zhang et al., 2018c; Merget et al., 2018), and regress human body keypoint (Tekin et al., 2017; Newell et al., 2016). While it is possible to output a separate heatmap per class (Zhang et al., 2018c; Tekin et al., 2017), most heatmap-based approaches do not distinguish between instances. Yi et al. (2016) re-formulate the problem based on each instance, but in doing so introduce a sharp difference between training and testing. Grids can also be used with heatmaps (Detone et al., 2018), but this results in an unrealistic assumption of uniformly distributed detections in the image. Overall, heatmap-based methods excel when the "final" task of the network is generate a heatmap (Merget et al., 2018), but are problematic to use as an intermediate layer in the presence of multiple instances.

**Sequential inference methods.** Another way to approach multi-instance problems is to attend to one instance at a time in a sequentially. Training neural network-based models with sequential attention is challenging, but approaches using policy gradient (Ba et al., 2015) and differentiable attention mechanisms (Gregor et al., 2015; Eslami et al., 2015) have achieved some success for images comprising *small* numbers of instances. However, Recurrent Neural Networks (RNN) often struggle to generalize to sequences longer than the ones encountered during training, and while recent results on inductive reasoning are promising (Gupta et al., 2018), their performance does not scale well when the number of instances is large.

**Knowledge transfer.** To overcome the acquisition cost of labelled training data, one can transfer knowledge from labeled to unlabeled dataset. For example, Inoue et al. (2018) train on a single instance dataset, and then attempt to generalize to multi-instance domains, while Uijlings et al. (2018) attempt to also transfer a multi-class proposal generator to the new domain. While knowledge transfer can be effective, it is highly desirable to devise unsupervised methods such as ours that do not depend on an additional dataset.

**Weakly supervised methods.** To further reduce the labeling effort, weakly supervised methods have also been proposed. Wan et al. (2018) learn how to detect multiple instances of a single object via region proposals and ROI pooling, while Tang et al. (2018) propose to use a hierarchical setup to refine their estimates. Gao et al. (2018) provides an additional supervision by specifying the number of instances in *each* class, while Oquab et al. (2015) and Zhang et al. (2018b) localize objects by looking at the network activation maps (Zhou et al., 2016; Selvaraju et al., 2017). Shen et al. (2018) introduce an adversarial setup, where detection boxes are supervised by distribution assumptions and

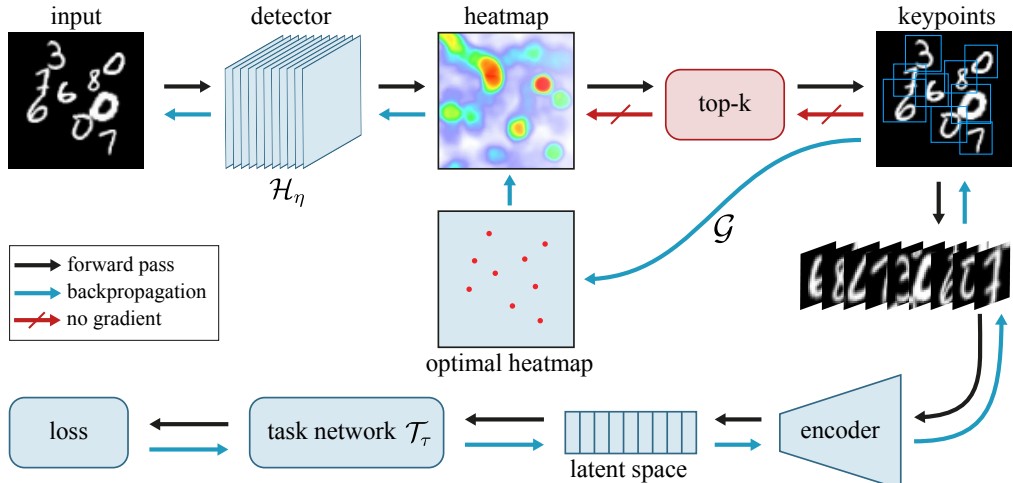

Figure 1: **The MIST architecture** – A network $\mathcal{H}_\eta$ estimates locations and scales of patches encoded in a heatmap $\mathbf{h}$. Patches are then extracted via a sampler $\mathcal{S}$, and then fed to a task-specific network $\mathcal{T}_\tau$. In this example, the specific task is to re-synthesize the image as a super-position of (unknown and locally supported) basis functions. As top-k operation is non-differentiable, we back-propagate by lifting through $\mathcal{G}$; see Section 3 for details.

classification objectives. However, all these methods still rely on region proposals from an existing method, or define them via a hand-tuned process.

## 3  MIST FRAMEWORK

We first explain our framework at a high level, and then detail each component in Section 4. A prototypical MIST architecture (Figure 1) is composed of two trainable components: ① the first module receives an image as input and extracts $K$ patches, at image locations and scales given by the top $K$ local maxima of a heatmap computed by a trainable heatmap network $\mathcal{H}_\eta$ with weights $\eta$. ② the second module processes each extracted patch with a task-specific network $\mathcal{T}_\tau$ whose weights $\tau$ are shared across patches, and further manipulates these signals to express a task-specific loss $\mathcal{L}_{task}$. The two modules are connected through non-maximum suppression on the scale-space heatmap output of $\mathcal{H}_\eta$, followed by a top-$K$ selection process to extract the parameters defining the patches, which we denote as $\mathcal{E}_K$. We then sample patches at these locations through (differentiable) bilinear sampling $\mathcal{S}$ and feed them the task module.

The defining characteristic of the MIST architecture is that they are *quasi-unsupervised*: the only supervision required is the number $K$ of patches to extract. However, as we show in Section 5.2, our method is not sensitive to the choice of $K$ during training. In addition, our extraction process uses a single heatmap for all instances that we extract. In contrast, existing heatmap-based methods (Eslami et al., 2015; Zhang et al., 2018c) typically rely on heatmaps dedicated to *each* instance, which is problematic when an image contains two instances of the same class. Conversely, we restrict the role of the heatmap network $\mathcal{H}_\eta$ to find the "important" areas in a given image, without having to distinguishing between classes, hence simplifying learning.

Formally, the training of the MIST architecture is summarized by:

$$\underset{\tau,\eta}{\text{minimize}} \ \mathcal{L}_{\text{task}}(\mathcal{T}_\tau(\mathcal{S}(\mathcal{E}_K(\mathcal{H}_\eta(\mathcal{I}))))) \tag{1}$$

where $\tau, \eta$ are the network trainable parameters. Unfortunately, the patch extractor $\mathcal{E}_K$ is non-differentiable, as it identifies the locations of the top-$K$ local maxima of a heatmap and then selects the corresponding patches from the input image. Differentiating this operation provides a gradient with respect to the input, but no gradient with respect to the heatmap. Although it is possible to smoothly relax the patch selection operation in the $K = 1$ case (Yi et al., 2016) (i.e. `argmax`), it is unclear how to generalize this approach to the case of *multiple* distinct local maxima. It is

---

**Algorithm 1** Multi-stage optimization for MISTs

---

**Require:** $K$ : number of patches to extract, $\mathcal{L}_{\text{task}}$ : task specific loss, $\mathcal{I}$ : input image, $\mathcal{G}$ : the keypoints to heatmap function, $\mathcal{H}$ : the heatmap network, $\eta$ : parameters of the heatmap network, $\mathcal{T}$ : the task network, $\tau$ : parameters of the task network, $\mathcal{E}_K$ : the top-$K$ operator.
1: **function** TRAINMIST($\mathcal{I}, \mathcal{L}_{\text{task}}$)
2:      **for** each training batch **do**
3:          $\tau \leftarrow$ Optimization step for $\mathcal{T}_\tau$ with $\mathcal{L}_{\text{task}} + \lambda\|\{\mathbf{x}_k\} - \mathcal{E}_K(\mathcal{H}_\eta(\mathcal{I}))\|_2^2$
4:          $\{\mathbf{x}_k\} \leftarrow$ Optimization step for $\{\mathbf{x}_k\}$ with $\mathcal{L}_{\text{task}} + \lambda\|\{\mathbf{x}_k\} - \mathcal{E}_K(\mathcal{H}_\eta(\mathcal{I}))\|_2^2$
5:          $\eta \leftarrow$ Optimization step for $\eta$ with $\|\mathcal{G}(\{\mathbf{x}_k\}) - \mathcal{H}_\eta(\mathcal{I})\|_2^2$
6:      **end for**
7: **end function**

---

thus impossible to train the patch selector parameters directly by backpropagation. We propose an alternative approach to training our model, via lifting.

**Differentiable top-K via lifting.** The introduction of auxiliary variables to simplify the structure of an optimization problem has proven effective in a range of domains ranging from non-rigid registration (Taylor et al., 2016), to robust optimization (Zach & Bournaoud, 2018). To simplify our training optimization, we start by decoupling the heatmap tensor from the optimization (1) by introducing the corresponding auxiliary variables $\mathbf{h}$, as well as the patch location variables $\{\mathbf{x}_k\}$ from the top-K extractor:

$$\underset{\eta,\tau,\mathbf{h},\{\mathbf{x}_k\}}{\text{minimize}} \ \mathcal{L}_{\text{task}}(\mathcal{T}_\tau(\mathcal{S}(\{\mathbf{x}_k\}))) \quad \text{s.t.} \quad \mathbf{h} = \mathcal{H}_\eta(\mathcal{I}), \quad \{\mathbf{x}_k\} = \mathcal{E}_K(\mathbf{h}). \tag{2}$$

We then relax the first constraint above to a least-squares penalty via a Lagrange multiplier $\lambda$:

$$\underset{\eta,\tau,\mathbf{h},\{\mathbf{x}_k\}}{\text{minimize}} \ \mathcal{L}_{\text{task}}(\mathcal{T}_\tau(\mathcal{S}(\{\mathbf{x}_k\}))) + \lambda\|\mathbf{h} - \mathcal{H}_\eta(\mathcal{I})\|_2^2 \quad \text{s.t.} \quad \{\mathbf{x}_k\} = \mathcal{E}_K(\mathbf{h}). \tag{3}$$

As in many methods that use keypoint supervision to regress heatmaps (Cao et al., 2018), we assume that a good heatmap generator $\mathcal{G}$ exists – that is $\{\mathbf{x}_k\} \approx \mathcal{E}_K(\mathcal{G}(\{\mathbf{x}_k\}))$ We can now rewrite our optimization as:

$$\underset{\eta,\tau,\mathbf{h},\{\mathbf{x}_k\}}{\text{minimize}} \ \mathcal{L}_{\text{task}}(\mathcal{T}_\tau(\mathcal{S}(\{\mathbf{x}_k\}))) + \lambda\|\mathbf{h} - \mathcal{H}_\eta(\mathcal{I})\|_2^2 \quad \text{s.t.} \quad \mathbf{h} = \mathcal{G}(\{\mathbf{x}_k\}). \tag{4}$$

We can now drop the auxiliary variable $\mathbf{h}$ and rewrite our optimization as:

$$\underset{\eta,\tau,\{\mathbf{x}_k\}}{\text{minimize}} \ \mathcal{L}_{\text{task}}(\mathcal{T}_\tau(\mathcal{S}(\{\mathbf{x}_k\}))) + \lambda\|\mathcal{G}(\{\mathbf{x}_k\}) - \mathcal{H}_\eta(\mathcal{I})\|_2^2, \tag{5}$$

and then approach the problem by *block coordinate descent* – where the energy terms not containing the variable being optimized are safely dropped, and we apply $\mathcal{E}_K$ to the penalty term of (5):

$$\underset{\tau,\{\mathbf{x}_k\}}{\text{minimize}} \ \mathcal{L}_{\text{task}}(\mathcal{T}_\tau(\mathcal{S}(\{\mathbf{x}_k\}))) + \lambda\|\{\mathbf{x}_k\} - \mathcal{E}_K(\mathcal{H}_\eta(\mathcal{I}))\|_2^2, \tag{6}$$

$$\underset{\eta}{\text{minimize}} \ \|\mathcal{G}(\{\mathbf{x}_k\}) - \mathcal{H}_\eta(\mathcal{I})\|_2^2. \tag{7}$$

To accelerate training, we further split (6) into two stages, and alternate between optimizing $\tau$ and $\{\mathbf{x}_k\}$. Being based on block coordinate descent, this process converges smoothly, as we show in Section D of the appendix. The summary for the three stage optimization procedure is outlined in Algorithm 1. Notice that we are not introducing *any* additional supervision signal that is tangent to the given task.

## 4   IMPLEMENTATION

### 4.1   COMPONENTS

**Multiscale heatmap network – $\mathcal{H}_\eta$.** Our multiscale heatmap network is inspired by LF-Net (Ono et al., 2018). We improve the network by applying modifications on how the scores are aggregated over scale. The network takes as input an image $\mathcal{I}$, and outputs multiple heatmaps $\mathbf{h}'_s$ of the same size for each scale level $s$. To limit the number of necessary scales, we use a discrete scale space with $S$ scales, and resolve intermediate scales via interpolation; see Section A of the appendix for details.

**Top-$K$ patch selection – $\mathcal{E}_K$.** To extract the top $K$ elements, we perform an addition cleanup through a standard non-maximum suppression. We then find the spatial locations of the top $K$ elements of this suppressed heatmap, denoting the spatial location of the $k^{\text{th}}$ element as $(x_k, y_k)$, which now reflect local maxima. For each location, we compute the corresponding scale by weighted first order moments (Suwajanakorn et al., 2018) where the weights are the responses in the corresponding heatmaps, *i.e.* $s_k = \sum_s \mathbf{h}'_s(x_k, y_k)s$

**Generating model for ideal heatmap – $\mathcal{G}(\{\mathbf{x}_k\})$.** For the generative model that turns keypoint and patch locations into heatmaps, we apply a simple model where the heatmap is zero everywhere else except the corresponding keypoint locations (patch center); see Section B of the appendix for details.

**Patch resampling – $\mathcal{S}$.** As a patch is uniquely parameterized by its location and scale, i.e. $\mathbf{x}_k = (x_k, y_k, s_k)$, we can then proceed to resample its corresponding tensor via bilinear interpolation (Jaderberg et al., 2015; Jiang et al., 2019) as $\{\mathbf{P}_k\} = \mathcal{S}(\mathcal{I}, \{\mathbf{x}_k\})$.

## 4.2 Task-specific networks

We now introduce two applications of the MIST framework. We use the *same* heatmap network and patch extractor for both applications, but the task-specific network and loss differ. We provide further details regarding the task-specific network architectures in Section C of the appendix.

**Image reconstruction / auto-encoding.** As illustrated in Figure 1, for image reconstruction we append our patch extraction network with a *shared* auto-encoder for each extracted patch. We can then train this network to *reconstruct* the original image by inverting the patch extraction process and minimizing the mean squared error between the input and the reconstructed image. Overall, the network is designed to *jointly* model and localize repeating structures in the input signal. Specifically, we introduce the generalized inverse sampling operation $\mathcal{S}^{-1}(\mathbf{P}_i, \mathbf{x}_i)$, which starts with an image of all zeros, and places the patch $\mathbf{P}_i$ at $\mathbf{x}_i$. We then sum all the images together to obtain the reconstructed image, optimizing the task loss

$$\mathcal{L}_{\text{task}} = \left\| \mathcal{I} - \sum_i \mathcal{S}^{-1}(\mathbf{P}_i, \mathbf{x}_i) \right\|_2^2. \tag{8}$$

**Multiple instance classification.** By appending a classification network to the patch extraction module, we can also perform multiple instance learning. For each extracted patch $\mathbf{P}_k$ we apply a shared classifier network to output $\hat{\mathbf{y}}_k \in \mathbb{R}^C$, where $C$ is the number of classes. In turn, these are then converted into probability estimates by the transformation $\hat{\mathbf{p}}_k = \text{softmax}(\hat{\mathbf{y}}_k)$. With $\mathbf{y}_l$ being the one-hot ground-truth labels of instance $l$ with unit norm, we define the multi-instance classification loss as

$$\mathcal{L}_{\text{task}} = \left\| \frac{1}{L} \sum_{l=1}^L \mathbf{y}_l - \frac{1}{K} \sum_{k=1}^K \hat{\mathbf{p}}_k \right\|_2^2, \tag{9}$$

where $L$ is the number of instances in the image. We empirically found this loss to be more stable compared to the cross-entropy loss, similar to Mao et al. (2017). Note here that we *do not* provide supervision about the localization of instances, yet the detector network will automatically learn how to localize the content with minimal supervision (i.e. the number of instances).

## 5 Results and evaluation

To demonstrate the effectiveness of our framework we evaluate the two different tasks. We first perform a quasi-unsupervised image reconstruction task, where *only* the total number of instances in the scene is provided, i.e. K is defined. We then show that our method can also be applied to weakly supervised multi-instance classification, where only image-level supervision is provided. Note that, unlike region proposal based methods, our localization network only relies on cues from the classifier, and *both* networks are trained from scratch.

### 5.1 Image reconstruction

From the MNIST dataset, we derive two different scenarios. In the *MNIST easy* dataset, we consider a simple setup where the *sorted* digits are confined to a perturbed *grid* layout; see Figure 2 (top).

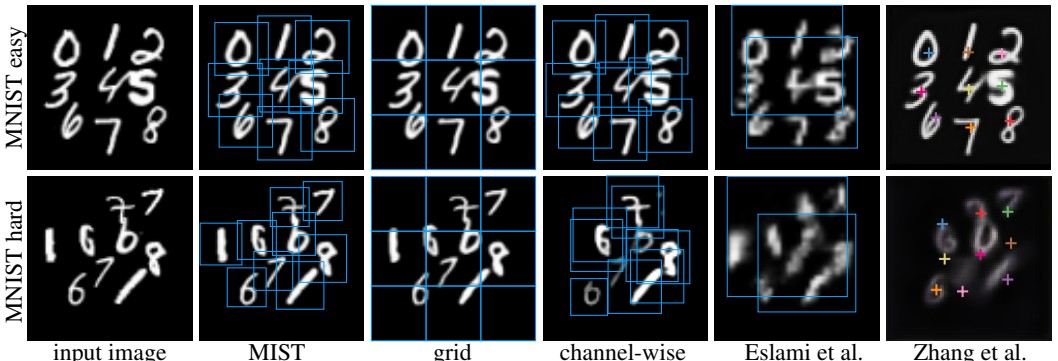

Figure 2: MNIST character synthesis examples for (top) the "easy" single instance setup and (bottom) the hard multi-instance setup. We compare the output of MISTs to grid, channel-wise, sequential Eslami *et al.* (Eslami et al., 2015) and Zhang *et al.* (Zhang et al., 2018c).

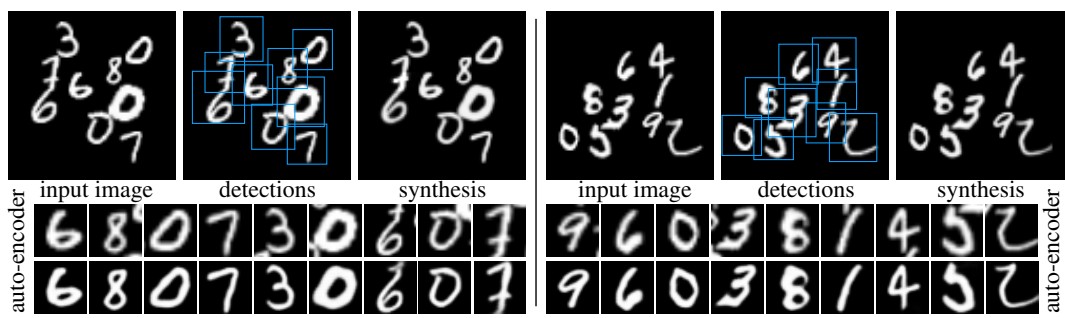

Figure 3: Two auto-encoding examples learnt from MNIST-hard. In the top row, for each example we visualize input, patch detections, and synthesis. In the bottom row we visualize each of the extracted patch, and how it is modified by the learnt auto-encoder.

Specifically, we perturb the digits with a Gaussian noise centered at each grid center, with a standard deviation that is equal to one-eighths of the grid width/height. In the *MNIST hard* dataset, the positions are randomized through a Poisson distribution (Bridson, 2007), as is the identity, and cardinality of each digit. We allow multiple instances of the same digit to appear in this variant. For both datasets, we construct both training and test sets, and the test set is never seen during training.

**Comparison baselines**  We compare our method against four baselines ① the *grid* method divides the image into a $3 \times 3$ grid and applies the same auto-encoder architecture as MIST to each grid location to reconstruct the input image; ② the *channel-wise* method uses the same auto-encoder network as MIST, but we modify the heatmap network to produce $K$ channels as output, where each channel is dedicated to an interest point. Locations are obtained through a channel-wise soft-argmax as in Zhang et al. (2018c); ③ *Esl16* (Eslami et al., 2015) is a sequential generative model; ④ *Zha18* (Zhang et al., 2018c) is a state-of-the-art heatmap-based method with channel-wise strategy for unsupervised learning of landmarks. For more details see Supplementary Section C.

**Results for "MNIST easy"**  As shown in Figure 2 (top) all methods successfully re-synthesize the image, with the exception of Eslami et al. (Eslami et al., 2015). As this method is sequential, with nine digits the sequential implementation simply becomes too difficult to optimize through. Note how this method only learns to describe the scene with a few large regions. We summarize quantitative results in Table 1.

**Results for "MNIST hard"**  As shown in Figure 2 (bottom), all baseline methods failed to properly represent the image. Only MIST succeeded at both localizing digits and reconstructing the original image. Although the grid method accurately reconstructs the image, it has no concept of individual digits. Conversely, as shown in Figure 3, our method generates accurate bounding boxes for digits even when these digits overlap, and does so without any location supervision. For quantitative results, please see Table 1.

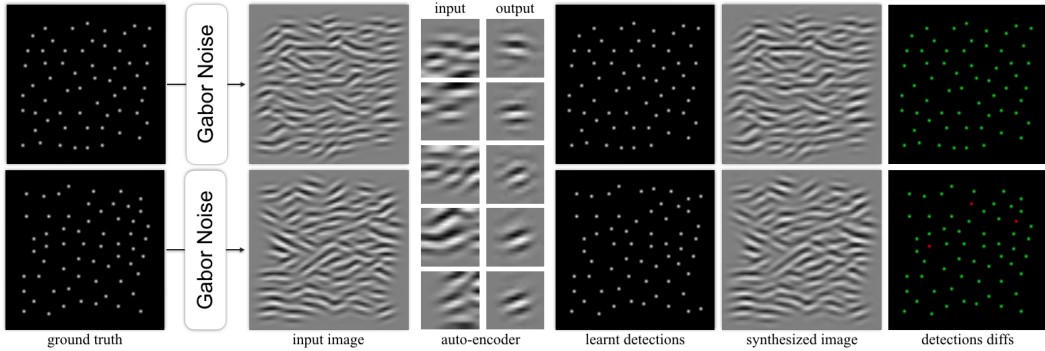

Figure 4: Inverse rendering of Gabor noise; we annotate correct / erroneous localizations.

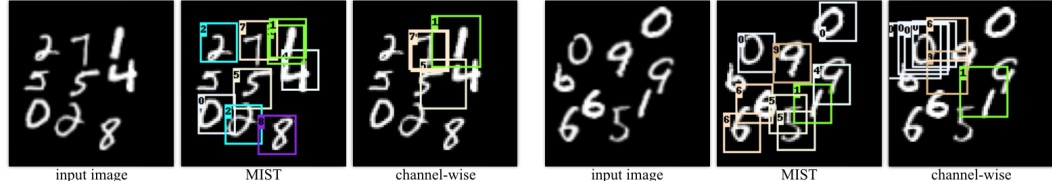

Figure 5: Two qualitative examples for detection and classification on our Multi-MNIST dataset.

**Finding the basis of a procedural texture** We further demonstrate that our methods can be used to find the basis function of a procedural texture. For this experiment we synthesize textures with procedural Gabor noise (Lagae et al., 2009). Gabor noise is obtained by convolving oriented Gabor wavelets with a Poisson impulse process. Hence, given exemplars of noise, our framework is tasked to regress the underlying impulse process and reconstruct the Gabor kernels so that when the two are convolved, we can reconstruct the original image. Figure 4 illustrates the results of our experiment. The auto-encoder learned to accurately reconstruct the Gabor kernels, even though in the training images they are heavily overlapped. These results show that MIST is capable of generating and reconstructing large numbers of instances per image, which is *intractable* with other approaches.

## 5.2 MULTIPLE INSTANCE CLASSIFICATION

**Multi-MNIST – Figure 5.** To test our method in a multiple instance classification setup, we rely on the *MNIST hard* dataset. We compare our method to *channel-wise*, as other baselines are designed for generative tasks. To evaluate the detection accuracy of the models, we compute the intersection over union (IoU) between the ground-truth bounding box and the detection results, and assign it as a match if the IoU score is over 50%. We report the number of correctly classified matches in Table 2, as well as the proportion of instances that are both correctly detected and correctly classified. Our method clearly outperforms the *channel-wise* strategy. Note that, even without localization supervision, our method correctly localizes digits. Conversely, the *channel-wise* strategy fails to learn. This is because *multiple instances* of the same digits are present in the image. For example, in the Figure 5 (right), we have three sixes, two zeros, and two nines. This prevents any of these digits from being detected/classified properly by a channel-wise approach.

**Sensitivity to $K$ – Table 3.** To investigate the sensitivity to the correctness of $K$ during training, we train with varying number of of instances and test with ground-truth $K$. For example, with $K = 6$,

| | MIST | Grid | Ch.-wise | Esl16 | Zha18 |
|---|---|---|---|---|---|
| MNIST easy | **.038** | .039 | .042 | .100 | .169 |
| MNIST hard | **.053** | .062 | .128 | .154 | .191 |
| Gabor | **.095** | - | - | - | - |

| | MIST | Ch.-wise | Supervised |
|---|---|---|---|
| IOU 50% | **97.8%** | 25.4% | 99.6% |
| Classif. | **98.8%** | 75.5% | 98.7% |
| Both | **97.5%** | 24.8% | 98.6% |

| Instances during train | AP$^{\mathrm{IoU=.50}}$ |
|---|---|
| $\{\mathbf{9}\}$ | **92.2%** |
| $\{6, \mathbf{7}, 8, 9\}$ | 90.1% |
| $\{3, 4, 5, \mathbf{6}, 7, 8, 9\}$ | 90.8% |

Table 1: Reconstruction error (root mean square error). Note that Grid *does not* learn any notion of digits.

Table 2: Instance level detection and classification performance on "MNIST hard".

Table 3: Sensitivity experiment of $K$ on the MNIST hard dataset.

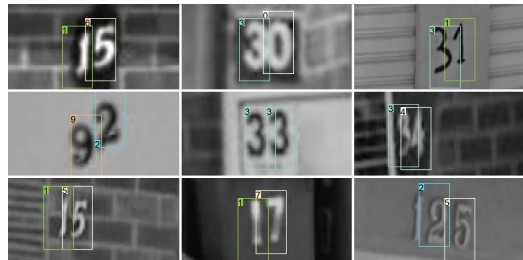

Figure 6: Qualitative SVHN results.

|  | MIST | Supervised |
|---|---|---|
| $AP^{IoU=.00}$ | **82.6**% | 65.6% |
| $AP^{IoU=.50}$ | **76.5**% | 62.8% |
| $AP^{IoU=.60}$ | **63.7**% | 59.8% |
| $AP^{IoU=.70}$ | 42.7% | **51.9**% |
| $AP^{IoU=.80}$ | 19.9% | **34.6**% |
| $AP^{IoU=.90}$ | 4.2% | **11.0**% |

Table 4: Quantitative SVHN results.

where the ground truth could be anything within $\{3, 4, 5, 6, 7, 8, 9\}$ – we mark the $K$ used during training with bold. Our method still is able to give accurate result with inaccurate $K$ during training. Knowing the exact number of objects is not a strict requirement at test time, as our detector generates a heatmap for the entire image regardless of the $K$ it was trained with. Note that while in theory sequential methods are free from this limitation, in practice they are able to deal with limited number of objects (e.g. up to three) due to their recurrent nature.

**SVHN – Figure 6 and Table 4.** We further apply MIST to the uncropped and unaligned Street View House Numbers dataset (Netzer et al., 2011). Compared to previous work that has used cropped and resized SVHN images (*e.g.* (Netzer et al., 2011; Ba et al., 2015; Goodfellow et al., 2014; Jaderberg et al., 2015)), this evaluating setting is significantly more challenging, because digits can appear anywhere in the image. We resize all images to $60 \times 240$, use only images labeled as containing 2 digits, and apply MIST at a single scale. Although the dataset provides bounding boxes for the digits, we ignore these bounding boxes and use only digit labels as supervision. During testing, we exclude images with small bounding boxes ($< 30$ pixels in height). We report results in terms of $AP^{IoU=.X}$, where X is the threshold for determining detection correctness. With IoU$= 0$, we refer to a "pure" classification task (i.e. no localization). As shown, supervised results provide better performance with higher thresholds, but MIST performs even better than the supervised baseline for moderate thresholds. We attribute this to the fact that, by providing direct supervision on the location, the training focuses too much on having high localization accuracy.

## 6    CONCLUSION AND FUTURE WORK

In this paper, we introduce the MIST framework for multi-instance image reconstruction/classification. Both these tasks are based on localized analysis of the image, yet we train the network without providing any localization supervision. The network learns how to extract patches on its own, and these patches are then fed to a task-specific network to realize an end goal. While at first glance the MIST framework might appear non-differentiable, we show how via lifting they can be effectively trained in an end-to-end fashion. We demonstrated the effectiveness of MIST by introducing a variant of the MNIST dataset, and demonstrating compelling performance in both reconstruction and classification. We also show how the network can be trained to reverse engineer a procedural texture synthesis process. MISTs are a first step towards the definition of optimizable image-decomposition networks that could be extended to a number of exciting *unsupervised* learning tasks. Amongst these, we intend to explore the applicability of MISTs to unsupervised detection/localization of objects, facial landmarks, and local feature learning.

Even though our method is a significant step forward from existing methods, it is not complete and has much potential for improvement. While our method is robust to $K$ during training as shown in Section 5.2, and $K$ can be determined dynamically during test time, it would beneficial to automatically determine $K$ through an additional network. Patches could also be refined to have varying aspect ratios similar to RPNs. Reconstruction could consider a global reconstructor to deal with complex images. These are parts of our future work towards MIST in the wild.

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

# Appendix

## A  MULTISCALE HEATMAP NETWORK

Our multiscale heatmap network is inspired by LF-Net (Ono et al., 2018). We employ a fully convolutional network with (shared) weights $\eta$ at multiple scales, indexed by $s = 1 \ldots S$, on the input image $\mathcal{I}$. The weights $\eta$ across scales are shared so that the network cannot implicitly favor a particular scale. To do so, we first downsample the image to each scale producing $\mathcal{I}_s$, execute the network $\mathcal{H}_\eta$ on it, and finally upsample to the original resolution. This process generates a multiscale heatmap tensor $\mathbf{h} = \{\mathbf{h}_s\}$ of size $H \times W \times S$ where $\mathbf{h}_s = \mathcal{H}_\eta(\mathcal{I}_s)$, and $H$ is the height of the image and $W$ is the width. For the convolutional network we use 4 ResNet blocks (He et al., 2015), where each block is composed of two $3 \times 3$ convolutions with 32 channels and relu activations without any downsampling. We then perform a *local spatial softmax* operator (Ono et al., 2018) with spatial extent of $15 \times 15$ to sharpen the responses. Then we further relate the scores across different scales by performing a "softmax pooling" operation over scale. Specifically, if we denote the heatmap tensor after local spatial softmax as $\tilde{\mathbf{h}} = \{\tilde{\mathbf{h}}_s\}$, since after the local spatial softmax $\mathcal{H}_\eta(\mathcal{I}_s)$ is already an "exponentiated" signal, we do a weighted normalization without an exponential, *i.e.* $\mathbf{h}' = \sum_s \tilde{\mathbf{h}}_s (\tilde{\mathbf{h}}_s / \sum_{s'} (\tilde{\mathbf{h}}_{s'} + \epsilon))$, where $\epsilon = 10^{-6}$ is added to prevent division by zero.

**Comparison to LF-Net.**  Note that differently from LF-Net (Ono et al., 2018), we do not perform a softmax along the scale dimension. The scale-wise softmax in LF-Net is problematic as the computation for a softmax function relies on the input to the softmax being *unbounded*. For example, in order for the softmax function to behave as a max function, due to exponentiation, it is necessary that one of the input value reaches infinity (i.e. the value that will correspond to the max), or that all other values to reach negative infinity. However, at the network stage where softmax is applied in (Ono et al., 2018), the score range from zero to one, effectively making the softmax behave similarly to averaging. Our formulation does not suffer from this drawback.

## B  GENERATIVE MODEL FOR THE HEATMAP

As discussed previously in Section 4.1, for the generative model that turns keypoint and patch locations into heatmaps, we apply a simple model where the heatmap is zero everywhere else except the corresponding keypoint locations (patch center). However, as the optimized patch parameters are no longer integer values, we need to quantize them with care. For the spatial locations we simply round to the nearest pixel, which at most creates a quantization error of half a pixel, which does not cause problems in practice. For scale however, simple nearest-neighbor assignment causes too much quantization error as our scale-space is sparsely sampled. We therefore assign values to the two nearest neighboring scales in a way that the center of mass would be the optimized scale value, making sure $\{\mathbf{x}_k\} = \mathcal{E}_K(\mathcal{G}(\{\mathbf{x}_k\}))$.

## C  IMPLEMENTATION DETAILS

**MIST auto-encoder network.**  The input layer of the autoencoder is $32 \times 32 \times C$ where $C$ is the number of color channels. We use 5 up/down-sampling levels. Each level is made of 3 standard non-bottleneck ResNet v1 blocks (He et al., 2015) and each ResNet block uses a number of channels that doubles after each downsampling step. ResNet blocks uses $3 \times 3$ convolutions of stride 1 with ReLU activation. For downsampling we use 2D max pooling with $2 \times 2$ stride and kernel. For upsampling we use 2D transposed convolutions with $2 \times 2$ stride and kernel. The output layer uses a sigmoid function, and we use layer normalization before each convolution layer.

**MIST classification network.**  We re-use the same architecture as encoder for first the task and append a dense layer to map the latent space to the score vector of our 10 digit classes.

**Baseline unsupervised reconstruction methods.**  To implement the *Eslami* et al. *(Eslami et al., 2015)* baseline, we use a publicly available reimplementation.[1] We note that Eslami *et al.* (Eslami

---

[1] `https://github.com/aakhundov/tf-attend-infer-repeat`

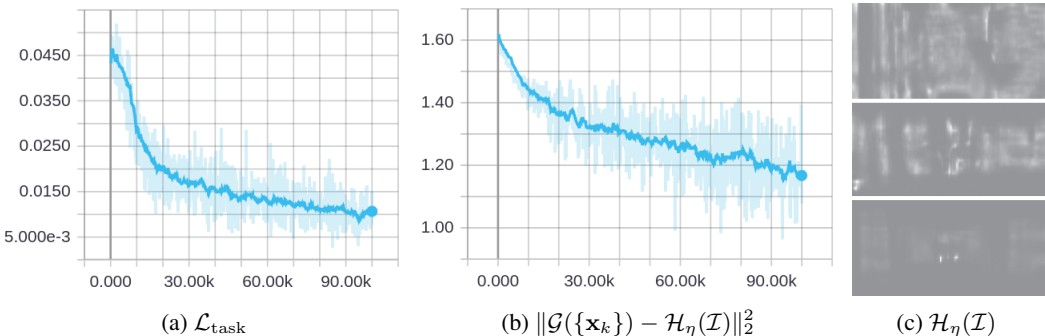

(a) $\mathcal{L}_{\text{task}}$         (b) $\|\mathcal{G}(\{\mathbf{x}_k\}) - \mathcal{H}_\eta(\mathcal{I})\|_2^2$         (c) $\mathcal{H}_\eta(\mathcal{I})$

Figure 7: Evolution of the loss and heatmap during training. (a) the task (classification) loss $\mathcal{L}_{\text{task}}$ and (b) the heatmap related loss $\|\mathcal{G}(\{\mathbf{x}_k\}) - \mathcal{H}_\eta(\mathcal{I})\|_2^2$ for each iteration. In (c), we show how the heatmap evolves for an example image from the SVHN dataset during training (from top to bottom) – brighter the intensity, higher the scores; see Section D.

## D    CONVERGENCE DURING TRAINING

As is typical for neural network training, our objective is non-convex and there is no guarantee that a local minimum found by gradient descent training is a global minimum. Empirically, however, the optimization process is stable, as shown in Figure 7. We reporpt both the training losses $\mathcal{L}_{\text{task}}$ and $\|\mathcal{G}(\{\mathbf{x}_k\}) - \mathcal{H}_\eta(\mathcal{I})\|_2^2$, and visualize how the heatmap evolves for an example image from the SVHN datset. We visualize the heatmap by showing brighter intensity values for higher scores. Each image is normalized separately with their own maximum and minimum values for better visualization. In Figure 7 (c) top, early in training, one can see that keypoints are detected at *random* locations as the heatmaps are generated by networks with randomly initialized weights. However, as training continues, keypoints that, by chance, land on locations nearby the correct object (e.g. numbers) for certain samples in a random batch, and become reinforced. Ultimately, as shown in (c) middle and bottom, MIST learns to detect these locations, thus learns to produce a peaky heatmap, and perform the task of interest. Note that this is unsurprising, as our formulation is a lifted version of this loss to allow gradient-based training.

## E    NON-UNIFORM DISTRIBUTIONS

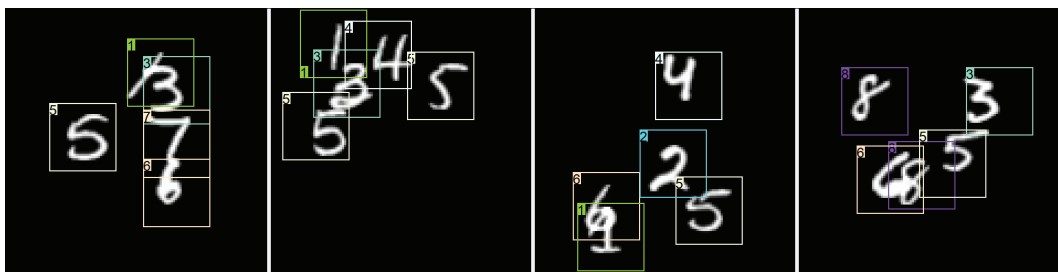

Figure 8: Examples with uneven distributions of digits.

Although the images we show in Figure 2 involve small displacements from a uniformly spaced grid, our method does not require the keypoints to be evenly spread. As shown in Figure 8, our method is able to successfully learn even when the digits are placed unevenly. Note that, as our detector is fully

convolutional and local, it cannot learn the absolute location of keypoints. In fact, we weakened the randomness of the locations for fairness against (Zhang et al., 2018c), which is not designed to deal with severe displacements.

