# OpenReview forum: "MIST: Multiple Instance Spatial Transformer Networks"
_ICLR.cc/2020/Conference — Reject_

### Official Review · AnonReviewer2 · 2019-10-07
**Official Blind Review #2**

**Rating:** 3

**Review:**

The article introduces the novel MIST architecture which tries to solve the problem of multiple-instance classification and image generation from multiple objects. It employs two submodels, where the first generates a heatmap of intereting region and the second model is a task-specific model that works on image-patches, for example a classifier or an autoencoder. Both models are connected by a patch-extraction routine. The main contribution of this paper is to provide a way to propagate errors through this non-differentiable patch-extraction scheme. This is done by introducing slack-variables.
------------------

The paper is overall relatively easy to follow and the results are very good. However, it suffers from the fact that it does not differentiate between model-architecture and the overall approach. While the main contribution is described in Section 4, the paper spends a lot of space beforehand to introduce the task-dependent models as well as the heatmap architecture - things that i can imagine will vary a lot in different applications. The real important part is how to train the model and this is unfortunately only half described. A good deal of abstraction from the network architecture would have made the paper a lot better.  Further, I think that the loss-function for the classification task does not work in the general case.


On my first read-through, i completely misunderstood Section 4. Here is an unsorted list of issues i had with this:
- since E_K is not truly invertible, writing the approximate inverse as E_K^{-1} is misleading.
- It might help to stress that you treat {x_k} as continuous and the sampling as differentiable.
- In (7) it would be better to explicitly write E_K^{-1}(x_k) instead of introducing \bar{h}. The line below is not clear.
- It is also misleading, because the choice of \bar{h}=E_K^{-1}(x_k) is not the minimizer of (5) given that all other variables are fixed. You can see this by observing that assuming that when {x_k}=E_K(H(I)) holds, we can choose all other pixels
to be exactly the value returned by H(I).
- I am not sure where the alternating part comes from because this usually involves taking your solution from (7) and feeding it into (6).
- I am pretty sure that in (6)+(7), as well as lines 3+6 of the algorithm, you actually don't want to optimize for tau or eta from scratch but only perform a single SGD step. I think that is what you are doing, but right now it is written as "find a complete new model for each batch".
- Since this is performed batch-wise: is x_k a variable kept between iterations or do you use H(I) for an initial estimate of x_k for the batch?

Regarding the classification objective:
-since (3) uses the MSE of the mean class-label and the mean-prediction, a dataset where all objects always appear with the exact same amount will not work since than for each image the mean label is identical.
- Therefore, the MNIST-easy dataset should be unsolvable for the proposed architecture since every digit occurs exactly once.


**Experience Assessment:**

I have read many papers in this area.

**Review Assessment: Checking Correctness Of Derivations And Theory:**

I assessed the sensibility of the derivations and theory.

**Review Assessment: Checking Correctness Of Experiments:**

I assessed the sensibility of the experiments.

**Review Assessment: Thoroughness In Paper Reading:**

I read the paper at least twice and used my best judgement in assessing the paper.

---

> ### Author Response · Authors · 2019-11-13
> **Response to Reviewer #2**
>
> Thanks for the great suggestions on how to improve the writeup. We have completely restructured the method presentation, significantly revised our derivation, and moved background work to the appendix. Please see the revised PDF for details, and let us know if you have further suggestions. For your convenience, we have colored the revised text with green.
>
> While it is true that the classification loss would fail when all classes are always present (e.g. MNIST_easy), this is true for all discriminative losses, e.g. hinge, cross-entropy. The suggested configuration of always having all classes -- numbers in MNIST -- equally present is equivalent to trying to learn a discriminative model with positive classes only. In other words, the answer to the classification is always going to be identical regardless of input. For example, when only pictures of cats are shown to a classifier when training to discriminate between cats and dogs, the classifier will learn to say cat regardless of the input. This is the limitation of a discriminative problem formulation, and not our loss function. Note that this is why we also show the reconstruction task, which does not suffer from this shortcoming.

---

### Official Review · AnonReviewer3 · 2019-10-27
**Official Blind Review #3**

**Rating:** 3

**Review:**

This paper proposes a method for multi-instance object classification and reconstruction that does not require any location-based supervision. The main contribution is the introduction of a differentiable top-K region proposal that allows to train the whole model with only a supervision of the total number of instances (and their class) in the image. They test the performance of their method in simple visual tasks like cluttered MNIST, street digit recognition and finding the basis of procedural texture generation.

The paper is well written and motivated. The proposed method is clear and well formalized. Their reported results seem substantially better than the baselines they compare against. The additional experiments in their Appendix Fig. 7, analyzing the evolution of the heatmap loss, is interesting, although I think the heatmap visualization could be improved to better understand what the model is learning across iterations (improve legend to include at what iteration the heatmap is taken, give on what image this is evaluated).

Unfortunately, their tasks seem quite easy, and it is hard to assess the impact of their method when working with more real-world data-sets, where the number of instances of every class is more loosely defined (we could always describe more objects in a real image from the COCO dataset for example). It seems of great importance to evaluate the limitations of their method in this direction, as the source of supervision might be too weak in the cases where the generated dataset might not have all the combinations of number of instances per image (as the cluttered MNIST has given that it’s procedurally generated). The results on SVHN are a little bit confusing, and it’s unclear what the “Supervised” method is, specially knowing that there are available methods that do obtain much better performance on this task using all the supervision available. It should also be better explained why the performance drops so drastically when the IoU threshold is increased. Finally, the texture generation experiments are very hard to interpret and are even further from real-world tasks.

Given my concerns on how this unsupervised approach can scale to real-life datasets, I suggest a weak reject, but I think the  proposed method has some interest for the community and I strongly encourage the authors to provide further evidence of performance of their method on more complex vision tasks.

**Experience Assessment:**

I have read many papers in this area.

**Review Assessment: Checking Correctness Of Derivations And Theory:**

I assessed the sensibility of the derivations and theory.

**Review Assessment: Checking Correctness Of Experiments:**

I carefully checked the experiments.

**Review Assessment: Thoroughness In Paper Reading:**

I read the paper at least twice and used my best judgement in assessing the paper.

---

> ### Author Response · Authors · 2019-11-13
> **Response to Reviewer #3**
>
> We have revised the caption of Fig. 7, and the associated text to clarify this experiment.
>
> Regarding the experiments, note that our method is a significant step forward over existing approaches. Without strong supervision, state of the art methods cannot cope with even such simple scenarios, and part of our contribution is pointing out these limitations. Applying our method to MS COCO is in our plans, but it is essential to identify the shortcomings of existing techniques before moving to large scale experiments. We show our method already has significant advances over prior approaches. Note that using synthetic images to explore novel frameworks/learning schemes is very common for learning literature. For example, a very recent work [1], also shares a similar goal of encoding image with parts, but is not real-world ready.
>
> SVHN EXPERIMENTS
> Notice our experiments are significantly more challenging than the ones found in existing works -- the reason for which the results are different. More specifically, most works use SVHN with cropped images using *known* bounding boxes. They use 32x32 pixel crops where individual digits are centered, e.g. [2,3]. Previous multi-digit approaches [4,5] used slightly larger crops, but still centered the crops around the known bounding boxes. In contrast, we *do not* use any localization supervision in our SVHN results to make the setup more realistic.
>
> TEXTURE EXPERIMENTS:
> The texture experiments are geared towards inverse graphics applications. Gabor filters and wavelets are a way of generating a procedural texture, and the experiment shows that the method is able to recover texture basis. Again, while not applied to real-world textures yet, our method is able to recover the required bases to synthesize a texture, opening up a new application area for deep learning in computer graphics.
>
> IoU THRESHOLD
> Regarding the steep decrease in accuracy that appears for high IoU threshold values, it is reasonable to expect this behavior in unsupervised learning. Since the network does not have to perfectly center the bounding boxes around the objects to accomplish its task successfully, it has no incentive to learn the boxes that align exactly with the ground truth. Also, the ground truth itself might be biased towards a certain way to align the boxes, which would explain why the supervised method does not suffer as much from a high IoU threshold.
>
> [1] Kosiorek A. R., Sabour S., Teh Y. W., and Hinton, G. E. (2019). Stacked Capsule Autoencoders. NeurIPS.
> [2] Goodfellow, I. J., Warde-Farley, D., Mirza, M., Courville, A., & Bengio, Y. (2013). Maxout networks. ICML.
> [3] Lin, M., Chen, Q., & Yan, S. (2013). Network in network. ICLR.
> [4] Goodfellow, I. J., Bulatov, Y., Ibarz, J., Arnoud, S., & Shet, V. (2013). Multi-digit number recognition from street view imagery using deep convolutional neural networks. arXiv preprint arXiv:1312.6082.
> [5] Jaderberg, M., Simonyan, K., & Zisserman, A. (2015). Spatial transformer networks. In Advances in neural information processing systems (pp. 2017-2025).

---

### Official Review · AnonReviewer1 · 2019-10-28
**Official Blind Review #1**

**Rating:** 6

**Review:**

The paper presents a way to solve a localization task for in images without providing any localization supervision. This allows tackling image reconstruction and classification problems that involve multiple object instances. At the core of the approach is a suggestion to resolve the non-differentiable top-k selection process: It is done by introducing axuilliary variables which allows the derivation of an alternating optimization procedure for their framework.

I like the paper and I think the problem it's tackeling is at the core for all many important image understanding tasks. When thinking about reasons that speak against the paper, I'm mostly concerned about the fact that the idea is not yet fully worked out:
* The biggest unresolved aspect is the number 'k' of instances needs to be provided. Assuming real-world relevant instance detection tasks, images in the test set never have 'k' provided.
* All patches need to be squared -- again, for real-world tasks this is not the case.
* In the case of image reconstruction, what is the 'reconstruction error' exactly? Assuming a scene with background clutter, how would this work here? It seems that the demonstrated examples have a black background so working additively is not a problem.

The conclusion mentions 'a first step', so I understand that the authors may be aware of the shortcomings. As such, I'd like to have limitations (and potential resolutions) to be pointed out in the paper.

With respect to related work, I'm missing "Attention-based Deep Multiple Instance Learning", by Ilse et al. I'm not certain if it could be applied to the reconstruction task, but it seems that it should be a baseline for the classification task. When talking about weak supervision, the paper "Weakly supervised object recognition with convolu-tional neural networks" by Oquab should also be mentioned in the related work section.

On a more detailed level, Figure 1 has no symbols for the newtork H_n and T_tau.

**Experience Assessment:**

I have read many papers in this area.

**Review Assessment: Checking Correctness Of Derivations And Theory:**

I assessed the sensibility of the derivations and theory.

**Review Assessment: Checking Correctness Of Experiments:**

I assessed the sensibility of the experiments.

**Review Assessment: Thoroughness In Paper Reading:**

I read the paper at least twice and used my best judgement in assessing the paper.

---

> ### Author Response · Authors · 2019-11-13
> **Response to Reviewer 1**
>
> We have added a thorough discussion on limitations to the conclusions, and updated Fig.1 as requested. For your convenience, we have colored the revised text with green.
>
> LIMITATIONS:
> 1) Regarding the number of K, while a limitation, we point out that it could be removed via an additional network that is specialized in determining K. Nonetheless, we show in our revised Section. 5.2 that, even with the wrong K, the network is able to learn. This is because we can dynamically change the number of K at test time. For example, learning with K=6, with a varying number of instances (3 to 9), and then testing with 9 gives only 2% performance degradation.
> 2) The patches are defined by location and scale for now. However, it is also possible to solve this in a way similar to region proposal nets, where an additional network is trained to refine bounding boxes.
> 3) Note the use of synthetic images to explore new architectures / training regimes is very common in the learning literature. For example [1] also aims to encode images with parts, but is not applicable to complex images yet.
>
> RELATED WORKS:
> 1) We have added “Attention-based Deep Multiple Instance Learning” and "Weakly supervised object recognition with convolutional neural networks" to the related work section.
> 2) About the comparison to “Attention-based Deep Multiple Instance Learning”, note this method requires that a set of potential candidate points to be given, and the method classifies given candidates in a more traditional multiple instance learning setup. The method performance would therefore depend highly on these candidates, which for our tasks it is non trivial to form. Therefore, we would like to note that it is not straightforward to form a proper pipeline using the method for our tasks *within the response period*.
>
> [1] Kosiorek A. R., Sabour S., Teh Y. W., and Hinton, G. E. (2019). Stacked Capsule Autoencoders. NeurIPS.

---

### Decision · Program_Chairs · 2019-12-19

**Decision:**

Reject

**Comment:**

Two reviewers are negative on this paper while the other one is slightly positive. Overall, this paper does not make the bar of ICLR. A reject is recommended.